# Harnessing Artificial Intelligence and Machine Learning for Identifying Quantitative Trait Loci (QTL) Associated with Seed Quality Traits in Crops

**DOI:** 10.3390/plants14111727

**Published:** 2025-06-05

**Authors:** My Abdelmajid Kassem

**Affiliations:** Plant Genomics and Bioinformatics Lab, Department of Biological and Forensic Sciences, Fayetteville State University, Fayetteville, NC 28301, USA; mkassem@uncfsu.edu

**Keywords:** artificial intelligence, machine learning, QTL mapping, seed quality, genomic prediction, deep learning, phenomics, multi-omics integration, feature selection

## Abstract

Seed quality traits, such as seed size, oil and protein content, mineral accumulation, and morphological characteristics, are crucial for enhancing crop productivity, nutritional value, and marketability. Traditional quantitative trait loci (QTL) mapping methods, such as linkage analysis and genome-wide association studies (GWAS), have played fundamental role in identifying loci associated with these complex traits. However, these approaches often struggle with high-dimensional genomic data, polygenic inheritance, and genotype-by-environment (GXE) interactions. Recent advances in artificial intelligence (AI) and machine learning (ML) provide powerful alternatives that enable more accurate trait prediction, robust marker-trait associations, and efficient feature selection. This review presents an integrated overview of AI/ML applications in QTL mapping and seed trait prediction, highlighting key methodologies such as LASSO regression, Random Forest, Gradient Boosting, ElasticNet, and deep learning techniques including convolutional neural networks (CNNs) and graph neural networks (GNNs). A case study on soybean seed mineral nutrients accumulation illustrates the effectiveness of ML models in identifying significant SNPs on chromosomes 8, 9, and 14. LASSO and ElasticNet consistently achieved superior predictive accuracy compared to tree-based models. Beyond soybean, AI/ML methods have enhanced QTL detection in wheat, lettuce, rice, and cotton, supporting trait dissection across diverse crop species. I also explored AI-driven integration of multi-omics data—genomics, transcriptomics, metabolomics, and phenomics—to improve resolution in QTL mapping. While challenges remain in terms of model interpretability, biological validation, and computational scalability, ongoing developments in explainable AI, multi-view learning, and high-throughput phenotyping offer promising avenues. This review underscores the transformative potential of AI in accelerating genomic-assisted breeding and developing high-quality, climate-resilient crop varieties.

## 1. Introduction

Seed quality traits play a fundamental role in crop production, influencing both agronomic performance and consumer preferences. These traits include seed size, oil content, protein composition, starch accumulation, germination rate, seed vigor, and longevity, all of which contribute to yield potential, nutritional value, and marketability. Improving seed quality is a major goal in crop breeding programs, as it directly impacts food security, industrial processing, and sustainable agriculture [1,2,3]. However, seed quality traits are typically controlled by multiple genes and are influenced by environmental factors, making their genetic dissection highly complex.

### 1.1. Traditional QTL Mapping and Its Limitations

Over the past few decades, significant progress has been made in identifying genetic loci associated with seed quality traits using quantitative trait loci (QTL) mapping and genome-wide association studies (GWAS). Traditional QTL mapping involves linkage analysis in biparental populations, allowing for the identification of genomic regions associated with phenotypic variation. While effective, this approach is often constrained by limited genetic diversity, low mapping resolution, and extensive time requirements for population development [4]. On the other hand, GWAS leverages natural genetic variation in diverse populations to detect marker-trait associations at a higher resolution [5]. However, GWAS is prone to false positives due to population structure and requires large sample sizes to achieve sufficient statistical power [6].

Despite the utility of these methods, traditional QTL mapping approaches face challenges in accurately predicting seed quality traits due to the polygenic nature of these traits, gene–environment interactions, and the complexity of underlying biological networks. The emergence of high-throughput genotyping and phenotyping technologies has led to an explosion of genomic and phenomic data, necessitating more advanced computational approaches to effectively analyze and interpret these datasets.

### 1.2. The Role of AI and ML in QTL Mapping

AI and ML have emerged as transformative tools in plant genomics, offering novel computational frameworks for handling high-dimensional data and improving QTL identification. ML algorithms, including deep learning, support vector machines (SVMs), random forests (RFs), and Bayesian networks, can efficiently process complex genomic datasets, uncover hidden patterns, and improve trait prediction accuracy [7,8]. Unlike traditional statistical models, ML approaches can capture nonlinear relationships between genetic markers and phenotypic traits, making them particularly well-suited for studying polygenic traits such as seed quality.

In recent years, AI and ML have been successfully applied in various aspects of crop breeding, including genomic selection, multi-omics data integration, and predictive modeling of agronomic traits [9,10]. These approaches enable the rapid identification of key genetic markers and provide insights into gene interactions and regulatory networks underlying seed quality traits. Additionally, AI-driven genomic selection models have demonstrated superior performance in predicting breeding values, allowing for more efficient selection of high-quality seed varieties in breeding programs.

Given the growing role of AI in plant genomics, this review aims to explore the application of AI and ML in identifying QTL associated with seed quality traits. First, it provides an overview of key seed quality traits and their genetic basis, emphasizing their importance in crop breeding and the challenges associated with their genetic dissection. Next, it discusses various AI and ML techniques used in QTL mapping and genomic prediction, highlighting how these approaches improve the accuracy and efficiency of trait identification compared to traditional methods. Furthermore, the review examines case studies that demonstrate AI-driven QTL discovery for seed quality traits, showcasing successful applications of ML models in different crop species. In addition, it addresses the challenges and limitations of AI-based QTL mapping, including issues related to data quality, model interpretability, computational complexity, and biological validation. Finally, it outlines future research directions and opportunities for integrating AI in crop breeding programs, focusing on emerging technologies and interdisciplinary collaborations that could further enhance the precision and applicability of AI in plant genomics. By synthesizing recent advancements in AI-driven QTL mapping, this review provides valuable insights into how AI can revolutionize the genetic improvement of seed quality traits, ultimately contributing to the development of high-yielding, high-quality crop varieties.

## 2. Seed Quality Traits and Their Genetic Basis

Seed quality traits encompass a wide range of physical, biochemical, and physiological characteristics that influence crop productivity and post-harvest quality. These traits can be broadly categorized into the following:**Physical Traits:** Seed size, shape, weight, and texture, which affect germination and processing quality [11].**Biochemical Traits:** Oil, protein, sugars, isoflavones, fatty acids, fiber contents, etc., which influence nutritional quality and industrial applications [12].**Physiological Traits:** Germination rate, seed vigor, dormancy, and longevity, which are critical for seed storage and crop establishment [13].

The genetic regulation of these traits is highly complex, often governed by multiple QTLs and influenced by environmental factors [12]. High-throughput phenotyping techniques, such as near-infrared spectroscopy (NIRS) and hyperspectral imaging, have enabled the precise measurement of seed quality traits, providing large datasets for AI-driven analyses [8,14,15]. Advances in genomics, including next-generation sequencing (NGS) and genotyping-by-sequencing (GBS), have further facilitated the identification of genetic markers associated with seed quality traits. Integrating AI and ML in this domain offers a powerful approach to deciphering complex genotype–phenotype relationships and enhancing the efficiency of marker-assisted breeding.

## 3. AI and ML Techniques for QTL Mapping

AI and ML are increasingly employed in plant genomics to uncover complex genotype–phenotype relationships. Unlike traditional statistical methods, ML models can handle high-dimensional, nonlinear data structures and are especially useful in capturing polygenic interactions, epistasis, and gene–environment interactions that underlie many seed quality traits.

### 3.1. Overview of AI and ML in Genomics

ML encompasses a range of algorithms capable of learning patterns from data to perform classification, regression, and clustering tasks. Several families of ML models have shown particular promise in plant QTL mapping (Table 1). These models have been applied to diverse tasks, from genomic prediction to trait classification and marker selection (Table 1).

### 3.2. Suitability of ML Methods for QTL Mapping Tasks

Depending on the research objective—trait prediction, classification, or marker discovery—different ML models are more appropriate.

#### 3.2.1. Feature Selection and Marker Prioritization

LASSO Regression and ElasticNet are favored due to their embedded feature selection ability. These models can identify key SNPs associated with target traits by shrinking irrelevant coefficients to zero. Random Forest also ranks features by importance but lacks the sparsity offered by LASSO.

#### 3.2.2. Trait Prediction and Genomic Selection

Gradient Boosting, Random Forest, and Support Vector Regression (SVR) have been used successfully in genomic prediction tasks where accuracy is prioritized over interpretability. Deep Learning models (e.g., DNNs, CNNs) have shown high performance, especially when integrated with phenomics or image-based trait data [7,14,16].

#### 3.2.3. Multi-Omics and Network-Based Integration

Graph Neural Networks (GNNs) and Bayesian networks are particularly suited for multi-layer data integration, enabling modeling of gene regulatory networks, metabolite–gene interactions, and more. For example, in a soybean case study [17], ElasticNet and LASSO outperformed tree-based models in predicting mineral accumulation traits, indicating their robustness in high-dimensional SNP data scenarios.

### 3.3. Practical Considerations for Model Selection

Dataset size: Tree-based and regularized linear models perform well on smaller datasets, while deep learning requires larger datasets.Computational resources: DL and ensemble models are computationally intensive and may require GPU infrastructure.Interpretability: Linear models and Random Forest offer more transparency, while deep models may require SHAP or LIME methods for explanation.

SHAP (SHapley Additive exPlanations) can show which SNPs or gene expression signals most influence trait predictions [18]. LIME (Local Interpretable Model-Agnostic Explanations) can help breeders understand why a specific SNP profile leads to a certain predicted trait outcome [19].

By leveraging these AI and ML techniques, researchers can accelerate the discovery of key genomic regions controlling seed quality traits and optimize breeding programs.

### 3.4. Hybrid and Ensemble Approaches for QTL Discovery

Recent advances have demonstrated that hybrid models—which combine multiple machine learning (ML) algorithms—can outperform single-method approaches in QTL prediction. For example, ensemble learning frameworks such as Stacked Generalization (Stacking) or Voting Classifiers have been applied to boost prediction accuracy by aggregating the strengths of multiple base learners (e.g., Random Forest, SVR, XGBoost) [8,20].

In QTL studies, these ensemble methods are especially valuable for integrating genomic, phenotypic, and environmental variables. For example, Khaki and Wang (2019) [21] demonstrated that a hybrid pipeline combining LASSO for feature selection and XGBoost for prediction significantly improved QTL-based trait prediction in soybean and maize breeding datasets. These hybrid frameworks also support cross-validation strategies that help minimize overfitting, which is a frequent challenge when working with small or medium-sized datasets in plant genetics [8].

### 3.5. Benchmarking AI Models in Trait Mapping

Quantitative benchmarking of ML models using real-world trial data is becoming increasingly common. In large-scale breeding programs for maize and wheat, studies have shown that ML models such as Random Forest (RF) and Support Vector Regression (SVR) often outperform traditional mixed linear models (MLMs) in predicting yield, protein content, and other polygenic traits, particularly under genotype-by-environment interactions [8,10]. In some cases, ML models achieved 10–20% higher prediction accuracy, especially in datasets with nonlinear structure or missing data points.

Such benchmark studies are essential to guide breeders in selecting model architectures appropriate for their data types and breeding objectives. Moreover, cross-environment validations using multi-environment trial (MET) data help quantify how well AI models generalize beyond training conditions—a key requirement for breeding applications [22].

## 4. Integrating Multi-Omics Data Using AI/ML for QTL Discovery

The complexity of seed quality traits—often governed by polygenic and genotype-by-environment interactions—necessitates an integrative approach to data analysis. Traditional single-omics analyses may miss key interactions occurring across different biological layers. Multi-omics integration, which combines data from genomics, transcriptomics, proteomics, metabolomics, and phenomics, provides a more holistic understanding of trait regulation. However, due to the volume and heterogeneity of such datasets, traditional statistical models are often insufficient. AI and ML techniques offer scalable, flexible frameworks for integrating and extracting meaningful insights from these high-dimensional, multi-model datasets. Figure 1 illustrates the AI/ML workflow for QTL mapping, demonstrating how genomic, phenotypic, and environmental data are processed through ML models to enhance QTL identification and genomic selection (Figure 1).

### 4.1. AI-Based Approaches for Multi-Omics Integration

Several AI/ML models have been adapted to handle multi-omics integration, each with unique strengths depending on the data structure and research objective. Multi-omics integration has been increasingly recognized as a transformative strategy in modern plant breeding, offering enhanced resolution in trait dissection and predictive modeling [23].

#### 4.1.1. Multi-View Learning

This approach treats each omics layer as a “view” of the data and learns joint representations that preserve both the individual structure of each omics layer and their shared relationships. In seed trait studies, gene expression (transcriptomics) and SNPs (genomics) can be modeled together to improve QTL resolution for nutrient-related traits. Algorithms like Multi-View SVM and canonical correlation analysis (CCA) variants are commonly applied.

#### 4.1.2. Graph Neural Networks (GNNs)

GNNs are designed to work on data represented as graphs, such as gene regulatory networks or metabolic pathways. Nodes (genes, SNPs, or proteins) are connected based on known biological relationships. GNNs can model SNP–gene–protein–phenotype interactions, allowing the identification of regulatory hubs influencing seed quality [24].

#### 4.1.3. Deep Neural Networks (DNNs) and Autoencoders

Deep learning models, including stacked autoencoders and variational autoencoders (VAEs), can compress high-dimensional multi-omics data into lower-dimensional latent representations for trait prediction or clustering. In rice, DNNs trained on integrated transcriptomic and metabolomic data improved starch trait predictions by >20% compared to single-omics models [8,14].

#### 4.1.4. Bayesian and Probabilistic Models

Bayesian networks offer a probabilistic framework to infer causal gene–trait relationships across omics layers. Predicting how variation in transcript levels influences metabolite abundance and seed morphology in cereals.

#### 4.1.5. Case Studies: Multi-Omics Integration in QTL Discovery

Recent studies highlight the power of multi-omics integration in enhancing the resolution and biological relevance of QTL identification (Table 2). These examples illustrate that multi-omics integration enables trait dissection at systems level, allowing breeders to move beyond association toward mechanistic insights (Table 2).

### 4.2. Challenges and Future Directions

While promising, multi-omics AI applications face several challenges:Data harmonization: Omics data may differ in scale, noise, or missing values.Model interpretability: Deep learning models require explainable AI add-ons.Experimental validation: Findings from integrated models must be functionally confirmed.

Future efforts should focus on developing standardized pipelines, incorporating explainability, and leveraging biological prior knowledge (e.g., gene ontologies, pathways) to guide integration.

### 4.3. Integration of Environmental and Epigenetic Layers

Beyond genomics and transcriptomics, epigenetic and environmental data layers are increasingly being integrated into QTL models to capture complex trait dynamics. Epigenomic features such as DNA methylation, histone modifications, and chromatin accessibility can be combined with SNPs and gene expression using multi-input neural networks, graph-based models, or multi-kernel learning techniques [25].

Environmental covariates—such as soil pH, temperature, rainfall, and radiation—have been shown to enhance predictive accuracy when incorporated alongside omics data. For instance, in rice and wheat, ML models that included both gene expression and environmental features outperformed transcriptomics-only models in predicting grain filling, seed dormancy, and yield under drought and heat stress [22,26]. These integrated models enable the discovery of G×E-sensitive QTLs, which are essential for breeding climate-resilient varieties.

### 4.4. Open-Source Platforms for Multi-Omics QTL Analysis

To support widespread adoption of AI in multi-omics analysis, several open-source platforms have been developed. Tools such as OmicsPipe [27], MOFA+ [28], and DeepMOCCA [29] provide scalable and modular frameworks for multi-omics data integration, particularly for large-scale QTL mapping and trait prediction.

These platforms offer features like automated preprocessing, dimensionality reduction, model training, and biological interpretation via embedded visualization tools. Their accessibility has helped democratize AI-assisted breeding, especially for programs with limited bioinformatics infrastructure. Moreover, they are increasingly being adapted to plant-specific workflows, supporting breeders in deploying omics-informed selection strategies at scale.

## 5. Case Studies on AI-Driven QTL Discovery

The application of AI and ML to QTL mapping is transforming our ability to identify loci associated with complex seed traits. Traditional methods such as linkage mapping and GWAS often fall short in resolving polygenic interactions, epistasis, and genotype-by-environment effects. AI/ML models offer significant improvements by enhancing prediction accuracy, enabling high-throughput marker discovery, and reducing computational bottlenecks [7,10,17].

Below, we summarize selected case studies that demonstrate these advantages, comparing AI-based outcomes with conventional methods wherever possible.

### 5.1. ML-Based QTL Mapping for Seed Mineral Nutrients in Soybean

Seed mineral content—such as nickel (Ni), molybdenum (Mo), iron (Fe), zinc (Zn), boron (B), and manganese (Mn)—is a key nutritional and agronomic trait. These elements exhibit complex inheritance patterns, often influenced by both genetic architecture and environmental factors.

Kassem (2025) [17] applied ML models to data from a Forrest × Williams 82 Recombinant Inbred Line (RIL) soybean population (*n* = 187), with 2075 SNP markers and phenotypic measurements collected from Carbondale, IL (2020). Models included LASSO Regression, ElasticNet, Random Forest, and Gradient Boosting (XGBoost). My results showed that LASSO and ElasticNet outperformed tree-based models, achieving higher R^2^ scores (up to 0.72 for Fe and 0.69 for Ni) and Lower RMSE across all traits (Figure 2). Random Forest and Gradient Boosting yielded negative R^2^ scores for Mo and Mn, indicating poor generalization and potential overfitting. ML methods identified key SNPs on chromosomes 8, 9, and 14, overlapping with previously reported QTLs [30,31]. Traditional QTL mapping (IM and CIM) detected the same loci but required larger sample sizes and failed to rank marker importance effectively. ML models provided prioritized, high-confidence SNPs and allowed cross-environmental prediction—not feasible in standard QTL pipelines.

### 5.2. ML Applications in Other Seed Quality Traits

ML and deep learning models have been successfully applied across a wide range of seed quality traits beyond mineral accumulation. These traits include seed morphology, oil and protein content, secondary metabolites, yield prediction, and image-based phenotyping. In many cases, AI/ML approaches outperform traditional regression or QTL mapping methods by modeling nonlinear relationships, improving prediction accuracy, and enabling high-throughput analysis.

#### 5.2.1. Seed Morphology and Weight Prediction

Image-based phenotyping has become a powerful tool in seed trait analysis. Seki and (2022) [32] used instance segmentation neural networks to extract seed shape traits and map 11 QTLs in lettuce, identifying loci linked to domestication with higher spatial resolution than GWAS alone. Similarly, Miranda et al. (2023) [33] applied convolutional neural networks (CNNs) to predict 100-seed weight (HSW) in soybean and barley, achieving 98% segmentation accuracy using RGB images. Duc et al. (2023) [34] reported even higher predictive accuracy for HSW using Random Forest and multiple linear regression (MLR) models (R^2^ = 0.98 and 0.94, respectively), surpassing traditional biometric methods.

#### 5.2.2. Oil and Protein Content

ML models such as support vector regression (SVR) and artificial neural networks (ANNs) have shown considerable promise in modeling complex biochemical traits. Yoosefzadeh-Najafabadi et al. (2023) [35] employed SVR-mediated GWAS to detect more significant loci for oil and protein content in soybean than the FarmCPU method. SVR effectively captured nonlinear and interactive effects that are often overlooked by linear mixed models.

In sesame, Abdipour et al. (2018) [36] demonstrated that ANN models outperformed MLR in predicting oil content under varied environmental conditions. Similarly, Parsaeian et al. (2020) [37] combined image analysis with ANN to estimate oil and protein content with high accuracy, showcasing the potential of vision-based AI tools for seed quality evaluation.

#### 5.2.3. Yield Prediction Across Crops

ANNs and other deep learning models have also been widely adopted for yield forecasting. In sesame, Emamgholizadeh et al. (2015) [38] used ANN models to predict seed yield more accurately than conventional regression techniques across environmental scenarios. In rapeseed, Niedbala (2019) [39] reported enhanced multi-criteria yield prediction using ANN compared to classical models. Likewise, Niedbala and Kozlowski (2019) [40] applied ANN for winter wheat yield forecasting, effectively integrating environmental variability.

Beyond yield, ML approaches have also been used to predict biochemical and food safety-related traits. For example, Niedbala et al. (2020) [41] used ANN to estimate ferulic acid, deoxynivalenol, and nivalenol concentrations in wheat grain with high accuracy—an important application in food safety.

Recurrent neural networks are also emerging in agricultural prediction. Haider et al. (2019) [42] used an LSTM neural network to forecast wheat production in Pakistan, demonstrating improved accuracy over conventional time series methods. In a more advanced application, Lee et al. (2019) [43] developed a self-predictable crop yield platform (SCYP) using deep learning that incorporated disease data to improve yield prediction under biotic stress conditions.

#### 5.2.4. Image-Based High-Throughput Phenotyping

Deep learning has revolutionized high-throughput trait extraction. Tang et al. (2024) [44] introduced GRABSEEDS, an automated CNN-based system for extracting plant organ traits from images, facilitating large-scale phenotyping across crops. Similarly, Sadeghi-Tehran et al. (2019) [45] developed DeepCount, a field-deployable CNN pipeline that accurately quantified wheat spikes in real time, improving yield estimation and reducing labor demands.

In sum, these diverse applications highlight the versatility and superiority of AI/ML tools over conventional methods for modeling and predicting seed quality traits. From yield forecasting to secondary metabolite modeling and image-based trait extraction, these tools are rapidly transforming both research and breeding practices across multiple crop species.

#### 5.2.5. Secondary Metabolites and Nutraceutical Traits

AI models have also proven effective for predicting secondary metabolite traits, which are crucial for nutritional, medicinal, and industrial applications but are often labor-intensive and costly to phenotype manually. These metabolites include essential oils, alkaloids, flavonoids, and terpenoids—traits influenced by complex genetic and environmental interactions.

For example, Ray et al. (2020) [46] applied artificial neural network (ANN) models to optimize coronarin D content in white ginger lily, also called garland flower (*Hedychium coronarium*), demonstrating how ANN-based optimization can help model metabolite accumulation under various conditions. Similarly, Niazian et al. (2018) [47] used both ANN and multiple regression analysis to predict essential oil content in ajowan (*Carum copticum*), showing higher predictive accuracy for ANN compared to classical methods.

These approaches offer non-destructive, high-throughput alternatives to traditional analytical chemistry methods and can accelerate the identification of high-performing genotypes in medicinal and aromatic plants. Moreover, the success of AI/ML in these systems highlights their versatility beyond staple crops, particularly in underutilized species where omics data may be limited but phenotyping bottlenecks are severe.

### 5.3. Comparative Summary of AI vs. Traditional Methods in Seed Trait QTL Studies

Table 3 provides a comparative overview of recent applications of AI and ML techniques versus conventional methods for seed trait prediction, phenotyping, and QTL discovery across a variety of crops. These case studies highlight the expanding role of AI not only in enhancing predictive performance, but also in enabling scalable, non-destructive, and high-throughput phenotyping [15].

Traditional methods such as manual phenotyping, linear regression, and classical statistical QTL mapping (e.g., linkage analysis and GWAS) have long served as the foundation for crop trait analysis. However, these approaches are often constrained by labor intensity, limited scalability, and an inability to capture nonlinear interactions or manage high-dimensional multi-omics data effectively [8,10].

By contrast, AI and ML approaches—including convolutional neural networks (CNNs), support vector regression (SVR), ensemble models, and deep learning architectures—have shown superior capabilities in modeling complex genotype–phenotype relationships and extracting meaningful patterns from large-scale datasets [7,14,15]. These methods also facilitate the integration of heterogeneous data sources, such as SNP markers, transcriptomics, image-based traits, and environmental data, thereby improving both QTL discovery and trait prediction.

For example, in soybean and barley, CNN-based image processing methods significantly outperformed manual phenotyping, achieving segmentation accuracies of up to 98% and reducing annotation requirements through synthetic datasets [33,51]. In tomato and rapeseed, deep learning models, including Mask R-CNN and Nu-SVR, led to substantial improvements in classifying seed quality and predicting yield, exceeding the performance of conventional regression models [52,53]. Additionally, tools like AIseed provide breeders with plug-and-play ML pipelines for automated seed trait analysis, reducing dependence on technical programming skills [54]. Tang et al. (2024) [44] developed GRABSEEDS, a deep learning-based platform for extracting plant organ traits from images, enabling precise and automated phenotyping across multiple species.

These findings collectively demonstrate that AI-driven approaches enhance prediction accuracy, operational efficiency, and biological insight, while also reducing human subjectivity and enabling deployment in real-time breeding scenarios. As summarized in Table 3, each case illustrates a clear improvement—whether in model performance, automation potential, or biological discovery—beyond what traditional methods typically offer.

## 6. Challenges and Limitations of AI/ML in QTL Mapping

Despite the growing success of artificial intelligence (AI) and machine learning (ML) in plant genomics, several technical, practical, and interpretative challenges continue to limit their widespread adoption in QTL mapping and breeding programs. Addressing these limitations is essential to ensure the reliability, scalability, and biological relevance of AI-powered approaches in crop improvement.

### 6.1. Data Quality, Quantity, and Availability

High-performing ML models rely heavily on large, high-quality datasets for training and validation. In plant breeding, however, datasets are often limited due to costly and labor-intensive phenotyping as well as inconsistent data collection across locations or seasons [8,10]. Multi-omics datasets—such as transcriptomics or metabolomics—may also suffer from missing values, batch effects, and noise, which reduce model robustness and generalizability.

Furthermore, most publicly available datasets are small relative to the complexity of the traits studied. This data scarcity can lead to overfitting and reduce a model’s ability to generalize across environments or populations. Improved efforts in data sharing, standardization, and cross-institutional collaboration are needed to address this bottleneck.

### 6.2. Model Interpretability and Biological Validation

A major barrier to the adoption of AI in genomics is the “black box” nature of many models—especially deep learning architectures. While these models may offer high predictive accuracy, they often lack interpretability, making it difficult for breeders and geneticists to extract actionable biological insights [8,14].

Tools such as SHAP, LIME, attention mechanisms, and saliency maps are being developed to improve interpretability. However, these tools are not yet standard in plant breeding pipelines. Additionally, even when ML models highlight significant features or SNPs, biological validation through wet-lab experiments—such as gene knockout, overexpression, or CRISPR editing—is still required to confirm causality.

### 6.3. Computational Complexity and Infrastructure Needs

Training AI models—especially on high-dimensional data from genomics, transcriptomics, and image-based phenotyping—can require substantial computational power, memory, and storage [24]. Deep learning models, in particular, may demand GPU acceleration, which is not always accessible to researchers in resource-limited settings.

While cloud-based platforms and high-performance computing (HPC) environments offer potential solutions, cost, technical expertise, and data privacy concerns remain major hurdles. Simplified ML toolkits and user-friendly interfaces will be essential to promote broader accessibility.

### 6.4. Ethical, Regulatory, and Equity Considerations

AI-driven plant genomics also raises ethical and regulatory questions, particularly around data ownership, intellectual property rights, and the equitable distribution of benefits. Large-scale genomic datasets are often generated by publicly funded institutions but may be used by private entities for commercial gain without fair benefit-sharing [16].

Moreover, the concentration of AI tools and resources in well-funded breeding programs may exacerbate disparities between high-resource and low-resource regions, limiting the global impact of genomic technologies. Establishing transparent governance frameworks, open-source tools, and inclusive research networks will be critical to ensure fairness in AI-assisted breeding.

### 6.5. Lack of Domain-Specific ML Expertise

Finally, the successful application of AI in plant genomics often requires interdisciplinary expertise spanning genetics, statistics, computer science, and plant biology. However, many breeding teams lack sufficient ML training, which can lead to inappropriate model selection, misinterpretation of results, or overreliance on default parameters. This highlights the need for capacity-building efforts, including training programs and collaborative platforms that bridge computational and biological sciences.

In summary, while AI/ML offers transformative opportunities for QTL mapping, a number of systemic and technical challenges must be addressed. Tackling these issues will require a multidisciplinary effort, improved infrastructure, open data sharing, and a commitment to model transparency and ethical implementation.

## 7. Future Directions and Opportunities

As artificial intelligence (AI) and machine learning (ML) technologies continue to evolve, their application in QTL mapping and plant breeding is expected to become more precise, interpretable, and integrated. Future research should focus on enhancing the accuracy, scalability, and transparency of AI models while promoting broader adoption across crops, environments, and institutions. Several key areas of opportunity are explored below.

### 7.1. Integration with Emerging Computational Technologies

Emerging tools such as quantum computing, edge computing, and federated learning present new opportunities for advancing plant genomics:Quantum computing could dramatically accelerate the analysis of large-scale multi-omics datasets by solving high-dimensional optimization problems in genotype–phenotype modeling [14];Edge computing can enable real-time genomic prediction and image-based phenotyping in the field using portable devices and sensors—especially relevant for low-resource or remote agricultural settings;Federated learning offers a privacy-preserving framework where institutions can train shared AI models without exchanging raw data, thus fostering cross-institutional collaboration.

These innovations may help address the computational bottlenecks and data-sharing limitations discussed in Section 6.

### 7.2. Development of AI-Assisted Breeding Pipelines

The next generation of breeding programs will benefit from AI-assisted decision support systems that integrate genotypic, phenotypic, and environmental data to guide selection decisions. These pipelines can automate key steps such as the following:Genomic selection and trait prioritization;Marker-assisted selection (MAS);Trait-environment optimization under climate change.

By incorporating predictive models early in the breeding cycle, researchers can reduce costs, shorten generation times, and increase selection efficiency [10,49].

### 7.3. Explainable and Interpretable AI Models

One of the most pressing needs in plant AI applications is the development of interpretable models that can generate biologically meaningful outputs. Future tools should perform the following:Integrate explainable AI (XAI) techniques such as SHAP values, feature attribution, and attention-based neural networks;Visualize gene-gene and marker-trait interactions using intuitive interfaces;Combine prior biological knowledge (e.g., gene ontologies, metabolic pathways) into model training.

This will help bridge the gap between computational predictions and biological validation, making AI models more actionable for plant scientists [8,14].

### 7.4. Open-Source Platforms and Equitable Access

To maximize the global impact of AI in crop improvement, future efforts should focus on developing open-access AI tools and user-friendly platforms for genomic prediction and QTL discovery. Examples include the following:Community-driven platforms like AIseed [54] for seed phenotyping;ML-based gene prioritization tools such as QTG-Finder2 [50];Shared ML pipelines, accessible via cloud-based workspaces.

Equally important is the commitment to training and capacity-building in low- and middle-income countries, ensuring that the benefits of AI-powered breeding are shared equitably across global agricultural systems [16].

## 8. Conclusions

The integration of AI and ML into plant genomics and QTL mapping is reshaping the landscape of crop improvement. These advanced computational techniques offer powerful alternatives to traditional methods by enabling accurate prediction of complex seed quality traits, robust marker-trait association analysis, and effective integration of multi-omics data.

This review has outlined how AI/ML models including LASSO, ElasticNet, Random Forests, Gradient Boosting, Support Vector Machines, Deep Neural Networks, and Convolutional Neural Networks are being applied across various crops to accelerate the discovery of QTLs for key traits such as seed weight, protein content, mineral accumulation, and yield components. In nearly all reported cases, AI methods have demonstrated superior performance over conventional approaches in terms of prediction accuracy, scalability, and biological insight.

Furthermore, we have emphasized the expanding role of AI in integrating high-dimensional multi-omics datasets, enabling systems-level understanding of trait architecture, and supporting high-throughput phenotyping platforms. At the same time, we have acknowledged key challenges related to data quality, model interpretability, computational resources, and equitable access to technology.

Looking ahead, the continued advancement of AI in plant breeding will depend on interdisciplinary collaboration, development of explainable and user-friendly tools, and a commitment to data sharing and capacity building. By embracing these innovations, the global plant science community can unlock new opportunities to enhance seed quality traits, accelerate breeding cycles, and contribute to resilient and sustainable agriculture.

## Figures and Tables

**Figure 1 plants-14-01727-f001:**
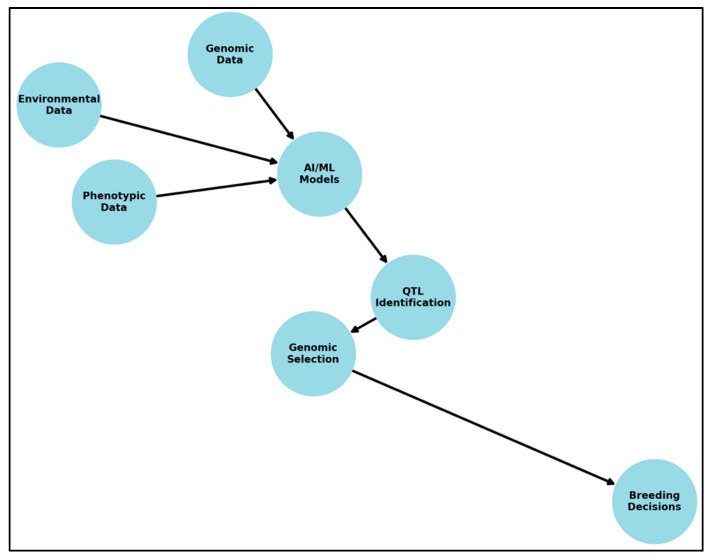
General workflow of artificial intelligence (AI) and machine learning (ML) applications in QTL mapping. The diagram illustrates key stages, including data preprocessing, feature selection, model training, validation, and biological interpretation for trait-associated loci.

**Figure 2 plants-14-01727-f002:**
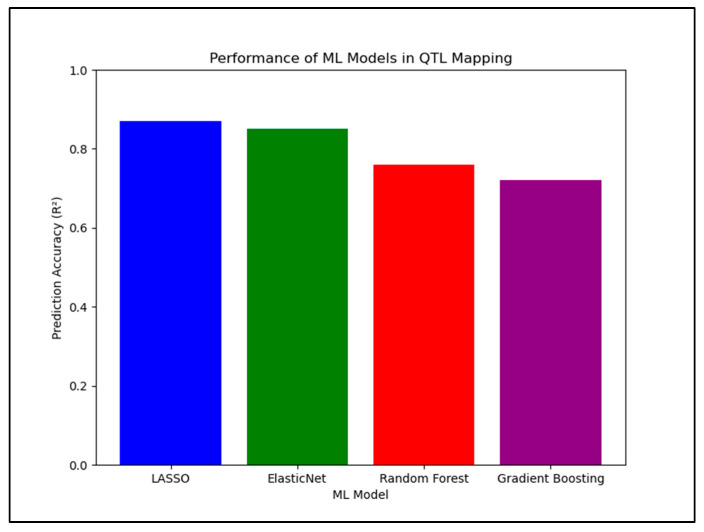
Comparative performance of different machine learning (ML) models used for QTL mapping of seed quality traits. Models are evaluated based on prediction accuracy (R^2^), root mean square error (RMSE), and generalization ability across traits and environments.

**Table 1 plants-14-01727-t001:** Comparison of commonly used machine learning (ML) techniques for QTL mapping. The table summarizes typical applications, strengths, and limitations of each model class, including regression-based, tree-based, kernel-based, and deep learning approaches.

ML Model	Main Use	Strengths	Limitations
LASSO Regression	Feature selection, SNP prioritization	Simple, interpretable; reduces overfitting	Assumes linear relationships
ElasticNet	Handling correlated features	Balances LASSO and Ridge regression benefits	Requires careful tuning
Random Forest (RF)	Classification, regression, SNP ranking	Nonlinear modeling, robust to noise	Prone to overfitting, less interpretable
Gradient Boosting (GB)	Trait prediction	High predictive accuracy	Sensitive to hyperparameters
Support Vector Machines (SVM)	Binary classification, regression	Good in high-dimensional spaces	Limited interpretability; slower training
Convolutional Neural Networks (CNNs)	Image-based phenotyping, seed shape	Learns hierarchical features	Requires large, labeled datasets
Deep Neural Networks (DNNs)	Multi-omics integration, trait prediction	Learns complex nonlinearities	Acts as a “black box”; high computational cost
Graph Neural Networks (GNNs)	Gene-gene or multi-omics network analysis	Captures topological interactions	Still emerging in plant sciences

**Table 2 plants-14-01727-t002:** Selected case studies highlighting the integration of multi-omics data using artificial intelligence (AI) and machine learning (ML) methods for enhanced QTL detection and trait prediction across major crops. Each example illustrates how combining genomic, transcriptomic, metabolomic, or proteomic layers improves model resolution and biological insight.

Crop	Integrated Omics	AI Technique	Key Outcome
Soybean	Genomics, Metabolomics	LASSO, DNNs	Improved detection of QTLs for oil and protein content
Rice	Genomics, Transcriptomics, Metabolomics	Deep Neural Networks	Accurate predictions of starch composition and amylose content
Wheat	Transcriptomics, Genomics	ML-Based Classification	Identified regulatory genes for dormancy and germination
Pigeonpea	Genomics, Proteomics, Metabolomics	Multi-layer ML pipeline	Discovered multi-trait QTLs for seed size, protein, and resistance

**Table 3 plants-14-01727-t003:** Comparative analysis of artificial intelligence (AI) and machine learning (ML) methods versus traditional statistical approaches for QTL mapping and trait prediction in seed quality studies. The table summarizes the crop and trait studied, the methods applied, and reported improvements in prediction accuracy, resolution, or efficiency.

Trait	Crop	Traditional Method	AI/ML Method	Reported Improvement	Reference
Mineral accumulation	Soybean	IM, CIM	LASSO, ElasticNet	↑ R^2^ (0.72 vs. ~0.4), ↓ RMSE	[17,30,31]
Oil/protein content	Soybean	FarmCPU-GWAS	SVR-GWAS	More QTLs detected; nonlinear relationships captured	[35]
Seed shape	Lettuce	Manual QTL mapping	CNN	11 QTLs linked to domestication; image-based accuracy improved	[32]
Seed yield	Cotton	eQTL only	XGBoost + eQTL	Identified pleiotropic yield genes (e.g., NF-YB3, GRDP1)	[48]
Seed weight (HSW)	Soybean	Multiple Linear Regression	Random Forest, MLR	R^2^: 0.98 (RF), 0.94 (MLR); higher than traditional MLR	[34]
Panicle traits	Rice	Manual phenotyping	Deep Learning	Real-time analysis of growth and yield-related traits	[49]
Protein and nutrient traits	Pigeonpea	Single-omics analysis	Multi-omics + ML	Multi-trait QTL detection; deeper functional insights	[15]
QTL gene discovery	Multi-crop	Statistical QTL mapping (e.g., GWAS, CIM)	ML-Based QTL Discovery	Machine learning improved QTL gene identification	[50]
Image-based HSW	Soybean	Manual image analysis	CNN + Image Processing	CNNs achieved 98% segmentation accuracy	[33]
Seed phenotyping	Barley	Manual annotation, classical morphometry	CNN + Synthetic Data	Synthetic data improved neural network training, high accuracy	[51]
Seed quality classification	Tomato	Visual inspection, lab germination tests	CNN + X-Ray Imaging	Mask R-CNN accurately classified seed viability and quality	[52]
Seed yield prediction	Rapeseed	Linear regression, ANOVA	Nu-SVR, MLPNN	Nu-SVR predicted seed yield with R^2^ = 0.86	[53]
Yield forecasting	Wheat	Time series regression models	LSTM Neural Network	Higher prediction accuracy than traditional forecasting methods	[42]
Image segmentation	Multi-crop	Thresholding, edge detection, manual masks	CNNs + Instance Segmentation	CNN-based methods highlighted as effective for plant phenotyping	[44]
High-throughput phenotyping	Multi-crop	Manual measurement, lab-based QC methods	AIseed Software + ML	Automated seed phenotyping and quality testing	[54]
Seed yield	Sesame	Linear regression models	Artificial Neural Network (ANN)	Higher accurate seed yield predictions across varied environments	[38]
Oil content	Sesame	Multiple Linear Regression	Artificial Neural Network (ANN)	ANN achieved higher prediction accuracy and lower error than MLR	[36]
Yield under disease stress	Multi-crop	Statistical yield forecasting	Deep Learning (SCYP)	Improved yield prediction by incorporating crop disease factors using deep learning	[43]
Essential oil prediction	Ajowan	Multiple Regression	ANN	ANN outperformed regression in modeling oil content	[47]
Coronarin D optimization	Garland flower	Manual lab-based optimization	ANN	Efficient prediction and optimization of metabolite content	[46]

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
