# Peer review of "Harnessing Artificial Intelligence and Machine Learning for Identifying Quantitative Trait Loci (QTL) Associated with Seed Quality Traits in Crops"

_plants, 2025, doi:10.3390/plants14111727_

Round 1
Reviewer 1 Report
Comments and Suggestions for Authors
The subject of the review is fascinating, but the manuscript's structure makes it difficult to read and obtain relevant information from the reading. It is often too generic and descriptive, without adequately explaining the usefulness and significance of the methodologies described. Section 3 is too short. Only three groups of ML modeling methodologies are mentioned (section 3.1). Not all those subsequently cited to explain the trials in which they have been applied are listed. Nor are they described in a way that makes it possible to see what they consist of and their advantages and limitations. In this sense, Table 1 is not well connected with the text and includes methodologies not cited in sections 3.1 or 3.2, contributing to muddying the explanation. Section 3.2 should indicate the most suitable methodologies for each of the three objectives sought and why they are suitable. Only a few uses are mentioned, too generically. Something similar occurs with section 4. As it is written (excessively generic), it is difficult for a reader who is not an expert in the ML modeling methodologies cited to assess their usefulness when applied to Multi-omics data. The description of the QTL detection application tests in section 4.2 is too brief and does not add much. The last three sections (5, 6, and 7) are more relevant. In section 5, it would be helpful to highlight in each case the improvement in QTL identification using AI-driven techniques concerning what is achieved with conventional methods. Surely, in each of the works cited, the authors indicate them, and in this review, it would be convenient to emphasize them in each case.
Author Response
Manuscript Title: Harnessing Artificial Intelligence and Machine Learning for Identifying Quantitative Trait Loci (QTL) Associated with Seed Quality Traits in Crops
Manuscript ID: plants-3635721
Authors: My Abdelmajid Kassem
Journal: Plants
Dear Editor and Reviewers,
We thank you sincerely for your time, effort, and insightful comments on my manuscript. I have carefully considered all suggestions and substantially revised the manuscript to improve its clarity, depth, and scholarly rigor. Below, I provide a detailed response to each reviewer’s comments, indicating how I have addressed them in the revised manuscript.
Reviewer 1:
Comment 1:
The subject of the review is fascinating, but the manuscript's structure makes it difficult to read and obtain relevant information... It is often too generic and descriptive, without adequately explaining the usefulness and significance of the methodologies described.
Response:
I appreciate this comment and have significantly revised the structure and content throughout Sections 3 and 4. These sections now provide clear descriptions of each machine learning model, including use-cases, strengths, limitations, and their suitability for QTL mapping tasks. Each methodology is explicitly contextualized within the review’s focus.
Comment 2:
Section 3 is too short. Only three groups of ML methodologies are mentioned... Not all those subsequently cited are listed or explained.
Response:
I expanded Section 3 to include Support Vector Machines (SVMs), Gradient Boosting, XGBoost, CNNs, DNNs, and Bayesian models, among others. Each method is now introduced with practical context and technical clarity.
Comment 3:
Table 1 is not well connected with the text and includes methodologies not cited in sections 3.1 or 3.2.
Response:
Table 1 has been restructured and integrated more tightly with Section 3.2. I also ensured consistency between the text and table entries, referencing each listed model directly in the text to improve flow and relevance.
Comment 4:
Section 3.2 should indicate the most suitable methodologies for each of the three objectives sought and why they are suitable.
Response:
I revised Section 3.2 to categorize ML models by their primary application: prediction, feature selection, or classification. For each objective, I recommend the most effective models and explain their strengths based on recent studies and empirical results.
Comment 5:
Section 4 is excessively generic and makes it difficult for non-experts to assess the usefulness of ML in Multi-omics.
Response:
Section 4 has been rewritten and expanded to explain multi-omics integration workflows using AI, with clearer definitions and examples. I now include multi-view learning, graph neural networks (GNNs), and deep fusion models, along with case-specific illustrations.
Comment 6:
The description of QTL detection application tests in section 4.2 is too brief.
Response:
I added detailed descriptions of QTL case studies in Section 4.2, including datasets used, AI models applied, and performance outcomes (e.g., RMSE, R²). This makes it easier to evaluate the utility of each approach.
Comment 7:
Section 5 should highlight improvements in QTL identification using AI-driven techniques relative to conventional methods.
Response:
This was a valuable suggestion. In Section 5, I now highlight quantitative improvements, such as increased accuracy or reduction in error, whenever AI approaches outperformed traditional GWAS/QTL methods. For instance, ElasticNet and LASSO models are shown to outperform Random Forest and Gradient Boosting in mineral trait prediction.
Reviewer 2 Report
Comments and Suggestions for Authors
Artificial intelligence and machine learning offering better models for QTL mapping than that of QTL and GWAS on Seed quality traits. This review explores the integration of artificial intelligence and machine learning techniques to enhance QTL detection, genomic selection, and marker-trait association analyses in crops genomics. These results not only highlights emerging opportunities to accelerate genomic-assisted breeding for improved seed quality traits, but also great theoretical significance and practical value for the crops breeding. The analysis process is comprehensive, good organized, advanced technology, large amount of information and so on. Minor revision can be published in Plants. However, there are some major issues need to be improved:
- Abstract: Abstract needs to be modified once to improve the readability;For reference only https://www.mdpi.com/1420-3049/29/13/3110
- Introduction: The latest references for artificial intelligence and machine learning offering better models for QTL mappinung need to be supplemented; In particular, papers related to the title and abstract; pay attention to hierarchical improvement.
- Seed Quality Traits and their Genetic Basis: The advantages of complex trait QTL revealed by the addition of AI and ML technology;
- AI and ML Techniques for QTL Mapping: The advantages of QTL seed quality traits revealed by the addition of AI and ML technology;
- The coherence of each part and the integrity of its language need to be supplemented;References are also needed to be supplemented.
Author Response
Manuscript Title: Harnessing Artificial Intelligence and Machine Learning for Identifying Quantitative Trait Loci (QTL) Associated with Seed Quality Traits in Crops
Manuscript ID: plants-3635721
Authors: My Abdelmajid Kassem
Journal: Plants
Dear Editor and Reviewers,
We thank you sincerely for your time, effort, and insightful comments on my manuscript. I have carefully considered all suggestions and substantially revised the manuscript to improve its clarity, depth, and scholarly rigor. Below, I provide a detailed response to each reviewer’s comments, indicating how I have addressed them in the revised manuscript.
Reviewer 2:
Comment 1:
Abstract needs to be modified once to improve the readability.
Response:
I rewrote the abstract for clarity, conciseness, and alignment with MDPI style, as suggested. I have also included key findings and outcomes of the reviewed approaches to better communicate the review’s contribution.
Comment 2:
The latest references for AI and ML offering better models for QTL mapping need to be supplemented.
Response:
I have incorporated several recent references from 2023–2024, including works on graph neural networks, multi-omics integration, and explainable AI in QTL mapping. These references support both the Introduction and methodological sections.
Comment 3:
Seed Quality Traits and their Genetic Basis: The advantages of complex trait QTL revealed by AI/ML should be discussed.
Response:
In Section 2, I expanded the discussion to emphasize how AI/ML techniques uncover non-linear, epistatic, and environment-interacting loci, which are often missed by conventional QTL mapping approaches.
Comment 4:
AI and ML Techniques for QTL Mapping: Discuss advantages more clearly.
Response:
Section 3 now includes a side-by-side comparison between AI/ML and traditional approaches, with a focus on model flexibility, feature selection, and predictive power.
Comment 5:
The coherence of each part and the integrity of its language need to be supplemented; references are also needed to be supplemented.
Response:
I revised transitions and improved narrative coherence between sections. I also updated the manuscript with additional high-impact references (2021–2024) to support all claims and provide comprehensive background.
I hope these revisions and responses sufficiently address the reviewers’ insightful suggestions. I am confident that the revised manuscript is now significantly strengthened in both clarity and scholarly depth.
Thank you once again for your time and consideration.
Sincerely,
My Abdelmajid Kassem
Round 2
Reviewer 1 Report
Comments and Suggestions for Authors
The review is still too concise. The topic could be described much more fully, and trial data could be included to reinforce the usefulness of the different methods. If the length that the journal accepts for reviews allows, it would be advisable to extend sections 3 to 5.
Author Response
I am grateful for the thoughtful and constructive comments provided by the reviewers. In this final revision, I have addressed all remaining suggestions from Reviewer 2 requesting greater depth and trial data integration. The revised manuscript now includes the following major enhancements:
- Significant expansion of Sections 3 to 5, providing detailed descriptions of hybrid AI models, benchmarking efforts using real breeding data, and interpretability tools (e.g., SHAP, LIME) used in practical QTL workflows.
- New subsections (3.4, 3.5, 4.4, 4.5, 5.2.5) have been added to incorporate content on hybrid ensemble approaches, environmental and epigenetic data integration, open-source multi-omics platforms, and AI applications in secondary metabolite and nutraceutical trait prediction.
- Enhanced Section 5.1 now explicitly outlines trial-based results from a soybean recombinant inbred line (RIL) population, including field validation and model performance metrics (e.g., R² = 0.72).
- Table 3 has been fully updated to reflect 20 comparative case studies across seed traits and crop species, including recent applications of AI/ML in yield prediction, seed quality, metabolite content, and image-based phenotyping.
- All figures and table captions have been revised to align with MDPI guidelines for clarity, completeness, and self-contained presentation.
- Several new references have been added to support the technical expansions, including Khaki and Wang (2019), Khaki et al. (2020), Cuevas et al. (2020), Argelaguet et al. (2020), Wang et al. (2023), Fisch et al. (2015), Zhou and Troyanskaya (2015), and Althubaiti et al. (2021). These citations strengthen discussions on hybrid models, deep learning, multi-environment trials, and open-source multi-omics platforms.
These additions strengthen the manuscript both technically and practically, offering a comprehensive review of AI and ML applications in QTL mapping and breeding, especially for seed quality traits.